# A behaviourally informed chatbot increases vaccination rates in Argentina more than a one-way reminder

**Dan Brown** [1] ✉, **Adelaida Barrera** [1] ✉, **Lucas Ibañez**[2], **Iván Budassi** [3], **Bridie Murphy**[1], **Pujen Shrestha**[1], **Sebastian Salomon-Ballada**[1], **Jorge Kriscovich**[4] **& Fernando Torrente** [5]

Maintaining COVID-19 vaccine demand was key to ending the global health emergency. To help do this, many governments used chatbots that provided personalized information guiding people on where, when and how to get vaccinated. We designed and tested a WhatsApp chatbot to understand whether two-way interactive messaging incorporating behaviourally informed functionalities could perform better than one-way message reminders. We ran a large-scale preregistered randomized controlled trial with 249,705 participants in Argentina, measuring vaccinations using Ministry of Health records. The behaviourally informed chatbot more than tripled COVID-19 vaccine uptake compared with the control group (a 1.6 percentage point increase (95% confidence interval, (1.36 pp, 1.77 pp)) and nearly doubled uptake compared with the one-way message reminder (a 1 percentage point increase (95% confidence interval, (0.83 pp, 1.17 pp)). Communications tools designed with behaviourally informed functionalities that simplify the vaccine user journey can increase vaccination more than traditional message reminders and may have applications to other health behaviours.

More than 13 billion COVID-19 vaccinations have been administered to date, averting tens of millions of deaths[1]. However, in Chaco, a disadvantaged province in northeast Argentina[2], only 33% of the eligible population had received a booster dose shortly before this project launched, 18 months into the vaccine rollout[3]. By way of comparison, in the city of Buenos Aires, 91% of residents had received a booster dose at this time[3]. Finding low-cost, scalable and effective interventions to increase vaccination is a key challenge that governments around the world are facing beyond COVID-19. Emerging technologies such as two-way chatbots are a promising avenue to address this challenge.

In this study, we developed a WhatsApp chatbot service to promote COVID-19 vaccine uptake in Chaco province. The interactive nature of the chatbot allowed us to incorporate a set of behavioural interventions to promote vaccination that are not feasible in a one-way message. Using a three-arm individual-level randomized controlled trial ($n = 249{,}705$), we evaluated the impact of the behaviourally informed chatbot on vaccination outcomes in comparison to both a control group and a single one-way reminder message.

Chatbots have become more prevalent in the past decade as the software that allows users to build and host them has become more readily accessible[4,5]. Recent advances in artificial intelligence have

[1]Behavioural Insights Team, London, UK. [2]ECOM Chaco S.A., Chaco, Argentina. [3]Unidad de Cienciares del Comportamiento y Políticas Públicas, Gobierno Federal de Argentina, Buenos Aires, Argentina. [4]Ministerio de Salud Pública de la Provincia de Chaco, Gobierno de la Provincia del Chaco, Chaco, Argentina. [5]Institute of Cognitive and Translational Neurosciences, CONICET, Universidad Favaloro and Fundación INECO, Buenos Aires, Argentina. ✉e-mail: dan.brown@cantab.net; adelaida.barrera@bi.team

**Fig. 1 | Images of the chatbot messages.** Chatbot messages providing eligibility information (left), helping users identify their nearest vaccine centres (middle) and reminding them the day before their vaccination (right). The images shown were modified from the originals to replace copyrighted material. Emojis adapted from OpenMoji under a Creative Commons licence CC BY-SA 4.0.

enabled the development of highly sophisticated and increasingly popular chatbot services. In customer service, chatbots are practically ubiquitous, whether used to triage enquiries before diverting to a human assistant or to complete whole tasks[6]. In health care, chatbots are still in an experimental phase. From the administrative to the ambitious, chatbots are currently being used to book health care appointments[7], provide diagnoses[8], encourage smoking cessation[9], administer cognitive behavioural therapy[10], monitor memory loss[11] and provide late-night companionship to people with insomnia[12], among other applications.

During the pandemic, more than 100 countries had a personalized chatbot providing information about COVID-19 symptoms, testing and other guidance to its population. At least 65 of these were produced by or on behalf of a national government (Supplementary Table 1). While there is a strong evidence base demonstrating that one-way text message reminders can increase vaccine uptake[13,14], no studies have tested whether an interactive chatbot incorporating a set of behavioural tools can do better.

We designed a chatbot to help ease the process of getting vaccinated using five core behavioural functionalities. First, users received individual-level personalized information on their eligibility based on the government's vaccinations database. Second, they could use a locator tool to find their nearest vaccination centres on the basis of their postcode or by using WhatsApp's 'Share Location' functionality. Third, they were prompted to make a plan for when and where they would get vaccinated. Fourth, they received a reminder message the day before their chosen date, and fifth, they were provided a link with Google Maps directions on how to get to their chosen centre. In each case, users could reply to the chatbot's messages by choosing an answer from a set menu of options. The screenshots in Fig. 1 show some of the functionalities, and the full details of the chatbot flow are available in Supplementary Table 2.

These features were designed to address a set of barriers to vaccine uptake that we identified through exploratory qualitative research among the local population in Chaco, including low visibility of when and where vaccines were available, a lack of direct government communications on eligibility, and out-of-date online information on vaccination centre locations and hours (see Supplementary Text for the details). The design of the features was based on existing evidence that suggests that making a behaviour easier by providing personalized information[15,16], prompting users to make a plan[17] and sending reminders[18] can encourage behaviour change.

Existing chatbot evaluations have largely focused on their acceptability to users and whether they successfully delivered the intended service within the chatbot itself, such as booking an appointment or administering a questionnaire[19–25]. Chatbots that try to encourage behaviour change have mostly been evaluated by measuring self-reported outcomes such as behavioural intentions and attitudes[26–29]. This includes one study in a laboratory setting showing that information provision in a chatbot may increase the self-reported intention to receive the COVID-19 vaccine[30]. A few randomized controlled trials have explored the effects of chatbots on physical activity using pedometer devices[31–34]. We address this important research gap by directly measuring the impact of the chatbot on the behaviour of interest: whether the individual got vaccinated.

To do so, we ran a preregistered three-arm randomized controlled trial with 249,705 participants in Chaco province[35]. Randomization was conducted at the individual level, with one third of the sample (83,235 participants) assigned to receive the WhatsApp chatbot, another third to receive a single one-way WhatsApp message encouraging them to get the next dose of the COVID-19 vaccine and the final third of the sample to receive no message (the control group).

Our primary outcome measure was a binary variable indicating whether the individual got their next dose of the COVID-19 vaccine

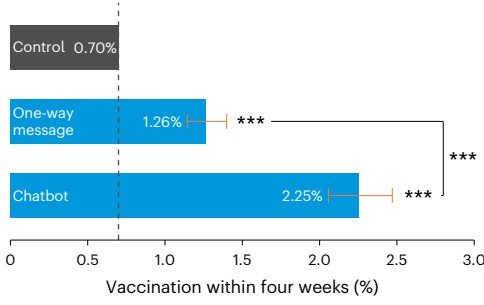

**Fig. 2 | The effect of the behaviourally informed chatbot on uptake of the next dose of the COVID-19 vaccine, compared with a one-way message and the control group.** Comparison of the vaccination rate in the control group against the vaccination rate we would expect if the control group received each treatment, given the average treatment effects we observed for each treatment arm. We used two-sided *z*-tests, adjusting *P* values using the Benjamini–Hochberg procedure to correct for all three between-arm comparisons. The asterisks (***) indicate significance at the 0.1% level. The *P* values for all three comparisons (chatbot versus control, one-way message versus control and chatbot versus one-way message) are <0.001, and the full results are reported in Table 1. The dashed black line represents the control-group vaccination rate. The orange bars represent 95% confidence intervals. Total *n* = 249,705 (83,235 in each trial arm).

within a four-week period. We used vaccination data from the Ministry of Health's NOMIVAC database (Registro Federal de Vacunación Nominalizado), provided by our partners at the Argentinian Ministry of Health.

By comparing vaccination rates between the chatbot trial arm and the control arm, we can understand the overall effect of the behaviourally informed chatbot on COVID-19 vaccination. By comparing the chatbot trial arm with the one-way message trial arm, we can understand whether a two-way interaction that incorporates behavioural functionalities causes a greater increase in vaccination rates than traditional one-way communications.

Participants for the trial were selected from one of three phone databases owned by the Ministry of Health (see Methods for the details). We restricted the sample to include individuals who had received at least one dose of the COVID-19 vaccine. Shortly before the launch of the trial, 83% of the Chaco population had received at least one dose of the vaccine, indicating that anti-vaccination sentiment was not widespread. However, uptake of the second dose was 70%, and only 33% had received a booster dose. Individuals who had not received any doses by this point in the pandemic were probably unwilling to get vaccinated, and so we believe they would not benefit from this chatbot service. The intervention intended to make the process of getting a vaccination easier for individuals who were not strongly vaccine-resistant. We also excluded any individual who was under the age of 18, had a mobile number that was not registered with WhatsApp, had a mobile number that was not unique to one individual in the databases, had more than one phone number across the databases or was not eligible for their next dose of the COVID-19 vaccine (see Supplementary Text for the details).

Our sample comprised 48.5% men and 51.5% women; 28.0% were between 18 and 29 years old, 47% were between 30 and 49 and 24.3% were 50 or older; 12.1% of participants had received one dose of the vaccine, 46.9% had received two doses, 40.3% had three doses and 0.6% had four. Compared with the adult population in Chaco, which is 47.6% male and 52.4% female, with 29.3% aged 18–29, 39.5% aged 30–49 and 31.2% aged 50 or older, our sample skews younger[36]. This is because older people were more likely to have completed their vaccination schedule when we started the trial and therefore did not meet the inclusion criteria.

**Table 1 | Results from the primary analysis**

| | Vaccinations within four weeks |
|---|---|
| One-way message | 0.598*** |
| | (<0.001)<br>(0.497, 0.700) |
| Chatbot | 1.187*** |
| | (<0.001)<br>(1.093, 1.280) |
| Observations | 249,705 |
| Control-group mean | 0.70% |
| Adjusted *P* value: one-way message versus chatbot | <0.001 |
| Estimated treatment effect of one-way message (pp) | 0.57 |
| Estimated treatment effect of chatbot (pp) | 1.56 |

This table reports the coefficients from a multivariate logistic regression where the dependent variable is a binary variable indicating whether the individual got their next dose of the COVID-19 vaccine within four weeks. We controlled for gender, age, the number of COVID-19 doses already received, time since the previous dose and the date of message delivery. We used two-sided *z*-tests and report adjusted *P* values that use the Benjamini–Hochberg procedure to correct for all three between-arm comparisons in parentheses. The *z*-values from the logistic regression (d.f. = 249,694) are equal to 11.533 for the one-way message arm and 24.856 for the chatbot arm. The estimated treatment effects in the final two rows are the effect at the control-group mean. The asterisks (***) indicate statistical significance at the 0.1% level. 95% confidence intervals are shown in parentheses.

## Results

### The chatbot increased the uptake of the COVID-19 vaccine

The behaviourally informed chatbot more than tripled COVID-19 vaccinations compared with the control group, who received no message (*P* < 0.001). As shown in Fig. 2, we observed a 1.56 percentage point increase (95% confidence interval, (1.36 pp, 1.77 pp)) in COVID-19 vaccinations in the chatbot group compared with the control group. Table 1 reports the coefficients from the underlying multivariate logistic regression, which controls for gender, age, the number of COVID-19 doses already received, time since the previous dose and the date of message delivery. The vaccination rate was balanced across all trial arms at baseline (Supplementary Table 3).

In absolute terms, 1,882 individuals in the chatbot group received their next vaccine dose in the four-week observation period, which is 1,300 more than in the control group. Had the chatbot been distributed across the full sample, it would have led to nearly 6,000 vaccinations. The chatbot was impactful across a wide range of demographics, with vaccination increasing for men and women of all age groups: 18–29, 30–49 and 50+ (Supplementary Table 4).

### The chatbot outperformed the one-way message reminder

The behaviourally informed chatbot nearly doubled vaccine uptake compared with the one-way message reminder. Vaccinations in the chatbot group increased by 0.99 percentage points (95% confidence interval, (0.83 pp, 1.17 pp); *P* < 0.001) compared with the one-way message. This effect amounted to an extra 824 vaccine doses in four weeks. The one-way message reminder also had a positive impact, causing a 0.57 percentage point increase in vaccine uptake compared with the control group (95% confidence interval, (0.44 pp, 0.70 pp); *P* < 0.001). The full results from the underlying logistic regression are reported in Table 1.

Our results show that while one-way message reminders increase vaccination, as documented extensively in previous research[13,14], governments can achieve nearly twice as large an impact by using a two-way interactive behaviourally informed chatbot. Furthermore, the chatbot significantly outperformed the one-way message reminders for men

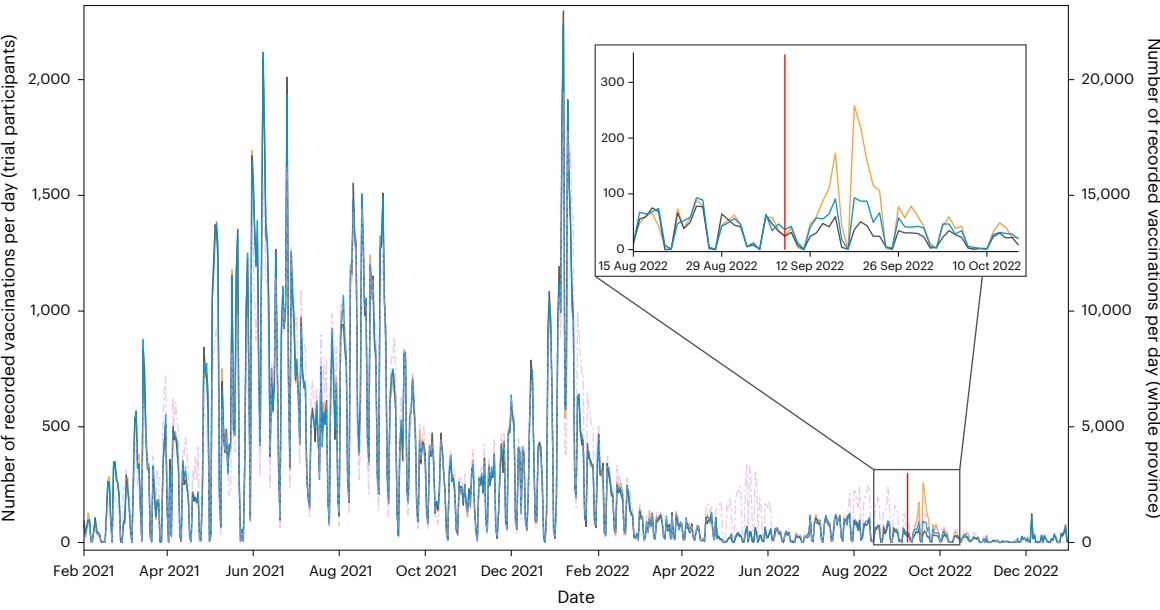

**Fig. 3 | The number of COVID-19 vaccine doses per day across the vaccine rollout in Chaco province.** The orange line represents the chatbot treatment group, the blue line represents the one-way message group and the grey line represents the control group (values shown on the left y axis). The dashed purple line represents the whole Chaco population (values shown on the right y axis). The vertical red line in the box indicates the date of the trial launch.

and women across all age groups: 18–29, 30–49 and 50+ (Supplementary Table 4). Unlike the one-way message, the chatbot offered both behaviourally informed functionalities and the option to communicate two-way. While we cannot test which of these aspects drove the effect of the chatbot, we know that the combination increased vaccination more than the traditional communications approach.

### Chatbots can be effective even with low vaccination demand

We launched the chatbot in September 2022 when COVID-19 infection rates were low. Demand for the COVID-19 vaccine was also low, as shown by the dashed purple line in Fig. 3, which represents the total number of COVID-19 vaccinations administered per day for the whole Chaco population since the start of the vaccine rollout in February 2021. The vaccination rate in our control group during the trial was just 0.70%. This is considerably lower than control groups in comparable studies, which typically have vaccination rates in the region of 5% to 70%[37–45].

Despite this, the chatbot increased vaccine uptake. The period enlarged in the upper right corner of Fig. 3 shows the sharp increase in vaccine uptake we observed for the chatbot group (orange line) shortly after the launch of our trial compared with the one-way message (blue line) and control (grey line) groups.

These findings suggest that our results may be valid for non-emergency contexts when demand is low. However, because the chatbot aims to make the vaccination process easier for those who are already motivated to do it, it is possible that the effects will be even larger in contexts when demand is high, such as the start of the flu season or the initial rollout of a new vaccine.

### User engagement was high but often incomplete

We achieved a high level of engagement with the chatbot, as 25% of users clicked through the initial message, and only 3.6% expressed a lack of interest. We monitored the proportion of users who interacted with each of the five core functionalities as outlined in Fig. 4.

Figure 4 demonstrates a key trade-off when designing the chatbot flow. Additional functionalities give people more intense support in navigating the vaccination user journey. But if the flow becomes too long, it risks disengagement. In future research, we will separately test the effects of different functionalities to help optimize chatbot design.

## Discussion

Two-way interactive chatbots allow governments to provide personalized support to individuals to make the process of getting a vaccination easier. Our study shows that by going beyond simple information provision, a behaviourally informed chatbot had nearly double the impact of a one-way message reminder on vaccine uptake.

The effect of the intervention is comparable to that of other 'nudges' or behaviourally informed interventions for adult vaccinations. Similar effect sizes on vaccine uptake (typically a few percentage points) have been found in studies offering small financial incentives[42] and prompting for implementation intentions[41] as well as those comparing behaviourally informed message frames to control groups who received no message[37–40]. Setting a default to book a vaccination appointment for all eligible individuals can have larger effects on vaccine uptake, closer to 10 percentage points[44,45]. However, in Chaco province there was no booking system in place at the time of our trial.

The behaviourally informed chatbot was effective even at a time when demand for vaccination was low. Risk perception and susceptibility to infection are important motivators to get vaccinated[46–48]. Identifying scalable interventions to sustain preventive behaviours when people are not motivated by a high-risk context is an important public health problem. Our study suggests that well-designed chatbots that incorporate behavioural functionalities can work and may be a promising tool to increase the uptake of routine vaccinations for governments around the world. They can also do so at low cost, as we estimate that the chatbot costs US$46 per additional vaccination. The advent of the R21 malaria vaccine, now recommended by the World Health Organization, further increases the potential to avert deaths by identifying effective demand-side vaccination strategies[49].

Beyond vaccination, chatbots of this kind could also have applications to almost any health behaviour that requires individuals to navigate an unfamiliar or effortful user journey, particularly behaviours that require meeting with health care professionals in person. Governments should nonetheless take care to ensure that users consider their chatbot trustworthy before implementing at scale, given evidence that chatbots can be considered less trustworthy than traditional information sources in some sectors (such as financial services)[50].

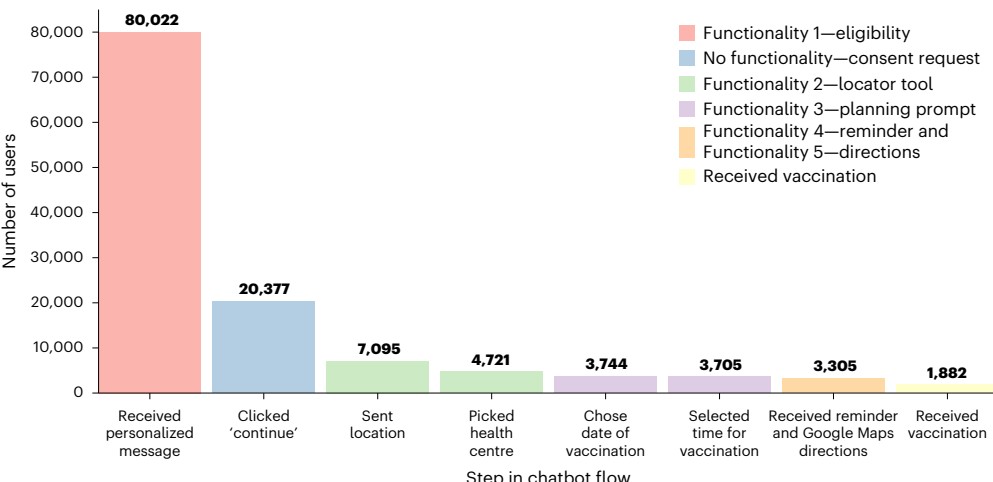

**Fig. 4 | User engagement with the chatbot.** The number of participants who completed each step in the chatbot flow out of the 83,235 assigned to the chatbot trial arm. The first bar represents the number of individuals who successfully received the first chatbot message with personalized eligibility information. Of the 3,705 individuals who completed the chatbot flow, 400 responded after

18:00 and chose the next day as their vaccination date. Since the reminder message script was executed each day at 18:00, these individuals did not receive a reminder message (which explains the decrease between the sixth and seventh bars, rather than a drop in engagement).

An important limitation of our study is that we do not know whether behaviourally informed chatbots can increase vaccine uptake in vaccine-resistant groups, as we excluded individuals who had not yet received their first dose of the COVID-19 vaccine from the sample. The chatbot was intended to make the process of getting a vaccination easier for those who already had at least some motivation to do so. It was not designed to address vaccine hesitancy or anti-vaccination attitudes. We are also unsure to what extent these results will generalize to contexts where demand for vaccinations is high, given that the chatbot launched at a time when perceived infection risk from COVID-19 was low.

Nevertheless, our three-arm randomized controlled trial has shown that chatbots embedded with behavioural tools can work and that they can work better than one-way message reminders. In the future, we hope to understand why that is the case. Testing different versions of the chatbot with different functionalities will allow us to provide recommendations on the mechanisms by which chatbots increase vaccine uptake and optimize their design.

## Methods

### Ethics
The study was assessed by an independent research ethics committee at Favaloro University Hospital (Comité de Bioética, Fundación Favaloro Hospital Universitario: CBE Acta No. 121, 29 June 2022) and approved prior to implementation. All procedures contributing to this work comply with the ethical standards of the committee. The committee accepted that the study would not seek participants' informed consent. Seeking informed consent would have been disproportionate to the interventions tested, would have been difficult to implement and would have potentially affected the validity of the studies' findings. The participants were not compensated for their participation. This research included local researchers (I.B., F.T. and L.I.) throughout the research process as indicated by the author contributions. The research was locally relevant and determined in collaboration with the Ministry of Health in Chaco province, as represented by J.K.

### Preregistration
Our study was preregistered at the AEA RCT Registry on 15 July 2022, which can be found at https://www.socialscienceregistry.org/trials/9758. We also obtained retrospective clinical trial registration on

7 June 2024 at the ISRCTN registry, which can be found at https://www.isrctn.com/ISRCTN13725439.

### Sample recruitment
We recruited adults in Chaco province, Argentina, who were eligible to receive their next dose of the COVID-19 vaccine. We first constructed a database of potential study participants using phone numbers from three administrative data sources provided by the Ministry of Health (see below). Each data source also contained individuals' national ID numbers (DNIs). From this database, we included any individual in the eligible sample who:

- Had received the first COVID-19 vaccine dose
- Was eligible to receive their next dose—that is, the second, third, fourth or fifth dose of the COVID-19 vaccine (see the Ministry of Health's vaccination eligibility criteria in the Supplementary Text)
- Was 18 years of age or older
- Had a mobile number registered with WhatsApp
- Had a mobile number that was unique to one individual in the study database
- Had only one mobile phone registered in the study database

Any individual who did not meet all these criteria was excluded. The data used to construct the eligible sample and conduct randomization were extracted on 22 August 2022. Our final sample comprised 249,705 individuals, representing all eligible participants. We conducted statistical power calculations assuming a significance level of 5%, a power of 80% and a two-sided test (with a more conservative Bonferroni correction for three comparisons), which gave us a minimum detectable effect size of 0.14 percentage points. For comparison, we reviewed the available literature investigating both SMS reminders for vaccines and the use of chatbots. This included a meta-analysis of 13 empirical studies that tested the effectiveness of digital interventions at increasing vaccine uptake[51] and a meta-analysis of 10 studies that tested the effectiveness of text message reminders on childhood vaccination[14].

### Data sources
We recruited our sample using the following data sources provided by the Ministry of Health. The data sources contained individuals' phone numbers and DNIs:

- Pasaporte Chaco: the Chaco province online services phone application, which contained a summary of personal health data related to COVID-19 and was mandatory for Chaco residents during lockdown periods
- 0800 helpline: a database from a phone helpline created by the Ministry of Health in Chaco to answer citizens' doubts on COVID-19 vaccinations and health services related to the pandemic
- SUMAR: the publicly subsidized Argentinian health care system, which provides health care to low-income citizens who are unable to contribute to social security

We obtained vaccination data from the government of Argentina's federal vaccination database (NOMIVAC). To our knowledge, this is a complete dataset of all COVID-19 vaccinations across the whole country. The data are uploaded to NOMIVAC directly by health care staff at vaccine centres through the online portal or mobile app from the Sistema integrado de información sanitaria, the national integrated health information system.

We merged data from the phone number data sources to the vaccinations database using individuals' DNIs—first, to identify the eligible sample on the basis of existing vaccinations, and, later, to construct our outcome measure. The anonymized dataset was transferred by the data controller following General Data Protection Regulation guidance.

### Randomization
The unit of randomization was the individual as identified by their DNI. Using a random number generator, we assigned the participants to one of three groups (chatbot group, one-way message reminder group and control group), stratifying by the number of COVID-19 vaccine doses they had already received (one, two, three or four). We did not expect participants to be aware that they were in a trial where different individuals were exposed to different conditions, meaning the participants were blind to the intervention. Data analysis was not performed blind to the conditions of the experiment.

### Construction of the primary outcome measure
Our primary outcome measure is a binary variable that takes the value 1 if an individual received their next dose of the COVID-19 vaccine within four weeks after they were enrolled into the trial according to the NOMIVAC database. One individual in the sample received more than one dose during the four-week follow-up period; for this individual, the outcome measure still took the value 1.

We counted the four-week follow-up period as the 28 days starting from the day after the participants were enrolled in the trial. We did this to be conservative, as the chatbot and one-way reminder messages were sent at 18:00 each day. It is likely that any vaccination logged for either group on the day of enrolment happened before they received the message.

### Statistical analysis
**Primary analysis of the impact on vaccination behaviour.** We conducted a multivariate logistic regression to estimate the intention-to-treat effect of the chatbot intervention on the binary primary outcome measure indicating whether the individual got vaccinated. We used the following regression:

$$Y_i \sim \text{Bernoulli}(p_i); \text{logit}(p_i) = \beta_0 + \beta_1 TC_i + \beta_2 TS_i + \mathbf{X}_i'\gamma$$

where the function logit is defined as the log-odds ratio, $\text{logit}(p) = \log(\frac{p}{1-p})$; $Y_i$ is a binary indicator of whether individual $i$ received a vaccination (1 if they did, 0 if not); $p_i$ is the probability that individual $i$ received a vaccination; $TC_i$ is a dummy variable indicating whether individual $i$ was assigned the chatbot (1 if they were, 0 if not); and $TS_i$ is a dummy variable indicating whether individual $i$ was assigned the one-way message reminder (1 if they were, 0 if not). $\mathbf{X}_i$ is a vector of pretreatment covariates: sex (binary), age as a categorical variable (18–29, 30–49 or 50+), the number of vaccine doses received (first dose, second dose, third dose or fourth dose), the length of time passed since the previous dose and the date of the initial intervention message.

We compared each treatment arm (chatbot and one-way reminder) to the control group. We also tested whether the chatbot performed significantly better than the one-way message reminder. We therefore made a total of three comparisons ($H_1$: $\beta_1 = 0$, $H_2$: $\beta_2 = 0$ and $H_3$: $\beta_1 = \beta_2$) and used the Benjamini–Hochberg procedure for multiple-comparison-adjusted $P$ values.

Partway through the implementation of the trial, the Chaco Ministry of Health requested that we not send messages to people who had already received four doses. As a result, a total of 240 individuals across the chatbot and one-way message reminder groups did not receive a message. To construct the primary outcome measure for these individuals (whether they received their next dose within 28 days of the message send date), we randomly assigned a message send date using the actual distribution of dates on which messages were sent for the chatbot and one-way message groups. Other than this issue, there were no missing values for any variable in the regression.

**Exploratory analysis of the impact on vaccination behaviour by number of previous doses and population subgroups.** We used a multivariate logistic regression to estimate the intention-to-treat effect of the chatbot intervention on the vaccination rate separately within each subgroup of participants who had received one, two or three doses before the launch of the trial and within different population subgroups for sex (male and female) and age (18–29, 30–45 and 50+). The exploratory analysis by dose was included in our prespecified analysis plan; the analyses by sex and by age were not. We estimated the following regression:

$$Y_i \sim \text{Bernoulli}(p_i); \text{logit}(p_i) = \beta_0 + \beta_1 TC_i + \beta_2 TS_i + \mathbf{X}_i'\gamma$$

where the function logit is defined as the log-odds ratio, $\text{logit}(p) = \log(\frac{p}{1-p})$; $Y_i$ is a binary indicator of whether individual $i$ received a vaccination (1 if they did, 0 if not); $p_i$ is the probability that individual $i$ received a vaccination; $TC_i$ is a dummy variable indicating whether individual $i$ was assigned the chatbot (1 if they were, 0 if not); and $TS_i$ is a dummy variable indicating whether individual $i$ was assigned the simple message reminder (1 if they were, 0 if not). $\mathbf{X}_i$ is a vector of pretreatment covariates: sex (binary), age as a categorical variable (18–29, 30–49 or 50+), the number of vaccine doses received (first dose, second dose, third dose or fourth dose), the length of time passed since the previous dose and the date of the initial intervention message. The covariates for number of doses, sex and age were removed when exploring the impact on vaccination behaviour of each respective covariate.

### Reporting summary
Further information on research design is available in the Nature Portfolio Reporting Summary linked to this article.

## Data availability
The data used in the analysis are publicly available via Dryad at https://doi.org/10.5061/dryad.31zcrjds1 (ref. 52). No accession code is required. The data are anonymity-preserving. The raw NOMIVAC dataset is restricted access and owned by the Ministry of Health. For information on how to request access to this dataset, please contact Iván Budassi at ivanbudassi@gmail.com.

## Code availability
The code used in the analysis is publicly available via Dryad at https://doi.org/10.5061/dryad.31zcrjds1 (ref. 52). No accession code is required. All data analysis was conducted using R version 4.4.0.

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

## Acknowledgements
We thank D. Perera, T. Hardy, M. W. Silva, S. Kettle and S. Merriam for their advice and support. We thank W. Giordano at ECOM, J. M. Castelli at the Federal Ministry of Health, C. Centeno at the Chaco Ministry of Health and her immunization and epidemiology teams for their support throughout the project. We also thank S. Grant and T. Hudacek at Global Impact for their encouragement. The chatbot accessed WhatsApp's Business Platform through the Business Service Provider Turn.io. This project was funded by the Vaccine Confidence Fund. F.T. received three grants from the Inter-American Development Bank, with grant numbers RG-T3106, 7200024037 and 7200027450.

The funders had no role in study design, data collection and analysis, decision to publish or preparation of the manuscript.

## Author contributions
Conceptualization: D.B., I.B. and F.T. Methodology: D.B., A.B., I.B., B.M., P.S. and F.T. Investigation: D.B., A.B., I.B., B.M., P.S., F.T. and S.S.-B. Formal analysis: P.S. Visualization: A.B. and P.S. Software: L.I. Project administration: D.B., A.B. and J.K. Funding acquisition: D.B., I.B., B.M. and F.T. Supervision: D.B., A.B., I.B. and F.T. Writing: D.B., A.B., B.M., P.S., S.S.-B. and F.T.

## Competing interests
The authors declare the following financial competing interests: D.B., A.B., B.M., P.S. and S.S.-B. were employed by the Behavioural Insights Team, L.I. was employed by ECOM, and F.T. was employed by INECO. I.B. was employed by the federal government of Argentina, and J.K. was employed by the Ministry of Health in Chaco province. The local government's role in the project was to provide access to the phone number databases from which participants were recruited for the trial, and to provide permission to implement the project.

## Additional information

**Correspondence and requests for materials** should be addressed to Dan Brown, Adelaida Barrera.

# Reporting Summary

## Statistics

For all statistical analyses, confirm that the following items are present in the figure legend, table legend, main text, or Methods section.

| n/a | Confirmed | |
|---|---|---|
| ☐ | ☒ | The exact sample size ($n$) for each experimental group/condition, given as a discrete number and unit of measurement |
| ☐ | ☒ | A statement on whether measurements were taken from distinct samples or whether the same sample was measured repeatedly |
| ☐ | ☒ | The statistical test(s) used AND whether they are one- or two-sided<br>*Only common tests should be described solely by name; describe more complex techniques in the Methods section.* |
| ☐ | ☒ | A description of all covariates tested |
| ☐ | ☒ | A description of any assumptions or corrections, such as tests of normality and adjustment for multiple comparisons |
| ☐ | ☒ | A full description of the statistical parameters including central tendency (e.g. means) or other basic estimates (e.g. regression coefficient) AND variation (e.g. standard deviation) or associated estimates of uncertainty (e.g. confidence intervals) |
| ☐ | ☒ | For null hypothesis testing, the test statistic (e.g. $F$, $t$, $r$) with confidence intervals, effect sizes, degrees of freedom and $P$ value noted<br>*Give P values as exact values whenever suitable.* |
| ☒ | ☐ | For Bayesian analysis, information on the choice of priors and Markov chain Monte Carlo settings |
| ☒ | ☐ | For hierarchical and complex designs, identification of the appropriate level for tests and full reporting of outcomes |
| ☒ | ☐ | Estimates of effect sizes (e.g. Cohen's $d$, Pearson's $r$), indicating how they were calculated |

*Our web collection on statistics for biologists contains articles on many of the points above.*

## Software and code

Policy information about availability of computer code

| | |
|---|---|
| Data collection | No software was used for data collection. The intervention was implemented using WhatsApp. |
| Data analysis | Data analysis was conducted using R version 4.4.0. |

For manuscripts utilizing custom algorithms or software that are central to the research but not yet described in published literature, software must be made available to editors and reviewers. We strongly encourage code deposition in a community repository (e.g. GitHub). See the Nature Portfolio guidelines for submitting code & software for further information.

## Data

Policy information about availability of data

All manuscripts must include a data availability statement. This statement should provide the following information, where applicable:
- Accession codes, unique identifiers, or web links for publicly available datasets
- A description of any restrictions on data availability
- For clinical datasets or third party data, please ensure that the statement adheres to our policy

We used data from the NOMIVAC vaccinations database (Registro Federal de Vacunación Nominalizado). The data used in the analysis are publicly available in the Dryad data repository at the following link: https://datadryad.org/stash/share/HPMMIkG8ltPniOJJChLa6ci81SaDKXK24I68g8njXCk. No accession code is required. The data are anonymity-preserving.

# Research involving human participants, their data, or biological material

Policy information about studies with <u>human participants or human data</u>. See also policy information about <u>sex, gender (identity/presentation), and sexual orientation</u> and <u>race, ethnicity and racism</u>.

| | |
|---|---|
| Reporting on sex and gender | Data on sex was used in the analysis. This data was collected from an administrative dataset. Of the 249,705 individuals in the sample, 48.5% were male and 51.5% female. Analysis was conducted by sex, as reported in Table S4 in the Supplementary Materials. The effect of both the one-way message and the chatbot treatment was statistically significant for both men and women: a 0.57ppt and 1.70ppt effect for men respectively, and a 0.57ppt and 1.42ppt effect for women respectively. |
| Reporting on race, ethnicity, or other socially relevant groupings | Data on race, ethnicity or other socially relevant groupings were not used in the study. |
| Population characteristics | In addition to data on sex, data on age and previous COVID-19 vaccinations was collected through administrative databases and used both as covariates for the regression analysis and for the sub-group analysis. Analysis by age and by previous number of COVID-19 vaccine doses is reported in Tables S4 and S6 in the Supplementary Materials respectively. |
| Recruitment | We recruited adults in Chaco province, Argentina, who were eligible to receive their next dose of the COVID-19 vaccine. Further details on eligibility criteria at the time of the study are outlined in the Supplementary Materials. To do this, we first constructed a database of potential study participants using phone numbers from three administrative data sources provided by the Ministry of Health (Pasaporte Chaco, SUMAR, and a 0800 helpline). Further details on each data source is provided in the Methods section. From this database, we included any individual in the eligible sample who: i) Had received the first COVID-19 vaccine dose, ii) Was eligible to receive their next dose; i.e., the 2nd, 3rd, 4th, or 5th dose of the COVID-19 vaccine, iii) Was 18 years of age or older, iv) Had a mobile number registered with WhatsApp, v) Had a mobile number that was unique to one individual within the study database, and vi) Had only one mobile phone registered within the study database. This provided us with a final sample of 249,705 participants. Given that we explicitly sampled individuals who had already received one dose of the COVID-19 vaccine at the time that the project launched, the results should not be interpreted as representative of the effect for the whole Chaco population (the effect of the chatbot could have been larger or smaller for people who had not yet received any doses). |
| Ethics oversight | The study was assessed by an independent research ethics committee at Favaloro University Hospital (Comité de Bioética, Fundación Favaloro Hospital Universitario: CBE Acta Nº 121, 29/06/2022) and approved prior to implementation. |

Note that full information on the approval of the study protocol must also be provided in the manuscript.

# Field-specific reporting

Please select the one below that is the best fit for your research. If you are not sure, read the appropriate sections before making your selection.

☐ Life sciences ☒ Behavioural & social sciences ☐ Ecological, evolutionary & environmental sciences

For a reference copy of the document with all sections, see <u>nature.com/documents/nr-reporting-summary-flat.pdf</u>

# Behavioural & social sciences study design

All studies must disclose on these points even when the disclosure is negative.

| | |
|---|---|
| Study description | Randomised controlled trial. The data is quantitative. |
| Research sample | Our research sample comprised adults (individuals aged 18 years and older) in Chaco province, Argentina, who were eligible to receive their next dose of the COVID-19 vaccine, and who met the other inclusion criteria described in the recruitment section above. As stated above, we restricted the sample to only include individuals who had received at least one dose of the COVID-19 vaccine. Individuals who had not received any doses by this point in the pandemic were likely to be unwilling to get vaccinated and so we believe they would not benefit from a chatbot service. However, shortly before the launch of the trial, 83% of the population in Chaco had received at least one dose of the vaccine. Of the 249,705 individuals in the sample, 48.5% were male and 51.5% female, 28.0% were between 18-29 years old, 47% were between 30-49 and 24.3% were 50 or older. The sample is not representative of the whole Chaco adult population: it does not include any individuals who had not yet received any doses of the COVID-19 vaccine, and it skews younger than the overall Chaco adult population (for whom the relevant proportions are: 47.6% male and 52.4% female, with 29.3% aged 18-29, 39.5% aged 30-49 and 31.2% aged 50 or older). |
| Sampling strategy | We included all individuals who met our eligibility criteria (described in the recruitment section above), which gave us a final sample size of 249,705 participants. Our sampling strategy was to include all eligible individuals. We conducted statistical power calculations based on this anticipated sample size, and compared the resulting minimum detectable effect size (approximately a 0.14 percentage point effect) to effect sizes of a comparable intervention (text message reminders) in the existing academic literature (which ranged from 0.7ppts to 7.4ppts) to ensure that the trial was adequately powered before we began. |
| Data collection | Data on our outcome measure was taken from an existing administrative dataset of all COVID-19 vaccinations, called NOMIVAC, which was provided to us by the Ministry of Health. We did not do any primary data collection for the outcome measure. The researchers were not blind to the experimental conditions or the study hypotheses. |

| Timing | Data for the primary outcome measure was taken from September 9th 2022 to October 19th 2022. The data collection period was extended by a further two weeks to 2nd November 2022 to create the outcome measure for one of the robustness checks. |
|---|---|
| Data exclusions | No data were excluded from the final sample of 249,705 participants described in the recruitment section above. |
| Non-participation | Our outcome measure is based on a complete administrative dataset of all vaccinations, and so there is no attrition from the sample. |
| Randomization | Participants were allocated randomly to the three experimental groups using a random number generator. Randomisation was stratified by the number of COVID-19 vaccine doses already received by the individual (1, 2, 3 or 4). |

# Reporting for specific materials, systems and methods

We require information from authors about some types of materials, experimental systems and methods used in many studies. Here, indicate whether each material, system or method listed is relevant to your study. If you are not sure if a list item applies to your research, read the appropriate section before selecting a response.

### Materials & experimental systems

| n/a | Involved in the study |
|---|---|
| ☒ | Antibodies |
| ☒ | Eukaryotic cell lines |
| ☒ | Palaeontology and archaeology |
| ☒ | Animals and other organisms |
| ☒ | Clinical data |
| ☒ | Dual use research of concern |
| ☒ | Plants |

### Methods

| n/a | Involved in the study |
|---|---|
| ☒ | ChIP-seq |
| ☒ | Flow cytometry |
| ☒ | MRI-based neuroimaging |

## Plants

| Seed stocks | *Report on the source of all seed stocks or other plant material used. If applicable, state the seed stock centre and catalogue number. If plant specimens were collected from the field, describe the collection location, date and sampling procedures.* |
|---|---|
| Novel plant genotypes | *Describe the methods by which all novel plant genotypes were produced. This includes those generated by transgenic approaches, gene editing, chemical/radiation-based mutagenesis and hybridization. For transgenic lines, describe the transformation method, the number of independent lines analyzed and the generation upon which experiments were performed. For gene-edited lines, describe the editor used, the endogenous sequence targeted for editing, the targeting guide RNA sequence (if applicable) and how the editor was applied.* |
| Authentication | *Describe any authentication procedures for each seed stock used or novel genotype generated. Describe any experiments used to assess the effect of a mutation and, where applicable, how potential secondary effects (e.g. second site T-DNA insertions, mosiacism, off-target gene editing) were examined.* |

