## [Peer Review File · Nature Human Behaviour]

Peer Review Information

Journal: Nature Human Behaviour

Manuscript Title: A behaviourally-informed chatbot increases vaccination rates in Argentina more than a one-way reminder

Corresponding author name(s): Dan Brown

Reviewer Comments & Decisions:

Decision Letter, initial version:

16th February 2024

Dear Dr. Brown,

Thank you once again for your manuscript, entitled "Increasing vaccination: reminders work, chatbots work better," and for your patience during the peer review process.

Your manuscript has now been evaluated by 3 reviewers, whose comments are included at the end of this letter. Although the reviewers find your work to be of interest, they also raise some important concerns. We are interested in the possibility of publishing your study in Nature Human Behaviour, but would like to consider your response to these concerns in the form of a revised manuscript before we make a decision on publication.

To guide the scope of the revisions, the editors discuss the referee reports in detail within the team, including with the chief editor, with a view to (1) identifying key priorities that should be addressed in revision and (2) overruling referee requests that are deemed beyond the scope of the current study. We hope that you will find the prioritised set of referee points to be useful when revising your study. Please do not hesitate to get in touch if you would like to discuss these issues further.

In particular, we ask that you address the following (as well as all other reviewer comments):

1) Address concerns raised by Reviewer 1 regarding the extent to which your study can attribute the findings to the effects of the chatbot itself. All interpretations must take into account the limitations of your study and be clear about what has been shown.

2) Address concerns highlighted by Reviewers 1 and 3 regarding the implications of the low response rate. Please carry out the additional sensitivity analyses recommended by Reviewer 1.

3) Ensure that the scope and limitations of your study are entirely clear.

In sum, we invite you to revise your manuscript taking into account all reviewer and editor comments. We are committed to providing a fair and constructive peer-review process. Do not hesitate to contact us if there are specific requests from the reviewers that you believe are technically impossible or unlikely to yield a meaningful outcome.

We hope to receive your revised manuscript within two months. I would be grateful if you could contact us as soon as possible if you foresee difficulties with meeting this target resubmission date.

- Include a "Response to the editors and reviewers" document detailing, point-by-point, how you addressed each editor and referee comment. If no action was taken to address a point, you must provide a compelling argument. When formatting this document, please respond to each reviewer comment individually, including the full text of the reviewer comment verbatim followed by your response to the individual point. This response will be used by the editors to evaluate your revision and sent back to the reviewers along with the revised manuscript.
- Highlight all changes made to your manuscript or provide us with a version that tracks changes.

[REDACTED]

We look forward to seeing the revised manuscript and thank you for the opportunity to review your work. Please do not hesitate to contact me if you have any questions or would like to discuss these revisions further.

Sincerely,

[REDACTED]

Reviewer expertise:

Reviewer #1: economics, policy studies in Latin America

Reviewer #2: communication, vaccine hesitancy

Reviewer #3: health behaviour, clinical trials

REVIEWER COMMENTS:

Reviewer #1:
Remarks to the Author:
A. Summary

The paper examines the impact of using interactive chatbots to provide personalized information to guide people to get vaccinated during a global health emergency. Using a pre-registered randomized controlled trial with 249,705 participants in Argentina, the authors can measure whether two-way interactive messaging can perform better than one-way message reminders. The authors found that the chatbot more than tripled COVID-19 vaccine uptake and nearly doubled uptake compared to the one-way message reminder.

B. Comments

1. Main comment.

The RCT is well done, and I commend the authors for their work and for working with the government of a relatively poor province in Argentina.

My main comment has to do with their interpretation of the results. Basically, the authors argue that "We found that the chatbot more than tripled COVID-19 vaccine uptake, and nearly doubled uptake compared to the one-way message reminder." However, is it true that the authors can separate the effect of the chatbot from the content of the information?

There seems to be the case that two things differ from one treatment to the other: the interface (that is, having the ability to select options or not) and the content of the information (that is, those using the chatbot received more and better-tailored information than the others). The authors do not provide evidence that would allow the reader to identify the driving source of the effects.

a. It could be the case that the one-way messages just provided less information, and that information was less behaviorally informed than the two-way messages. Therefore, it could be the case that the effect is not of the chatbot of itself but of the combination of behaviorally informed interventions that were delivered through the chatbot. In other words, it could be the case that users assigned to a one-way message receiving the same information would had acted the same.

b. Alternatively, it could well be that the option to click on the message is what drives people because they feel empowered, because they increase their trust in the government by seeing a more developed communication method, or because the development of the chatbot serves as a signal of how relevant vaccination may be.

This is a limitation of the RCT that the authors should recognize across the whole document -even perhaps by changing the title. In other words, the evidence indicates that a chatbot that helped to deliver a set of behaviorally informed messages increased vaccination. Whether it was the chatbot or the messages can't be identified with this exercise.

Relatedly, the authors also argue that "The effect of the chatbot is comparable to that of other 'nudges' or behaviorally informed interventions for adult vaccinations." Again, the chatbot is a mechanism to deliver a behaviorally informed intervention. The chatbot allows the authors to personalize the message, to have a call for action, and to send a reminder. These are all standard 'nudges' that could have been introduced through alternative technologies.

2. Policy recommendations

The overall change in the interpretation of the results has a bearing on the policy recommendations. They should reconsider and qualify sentences like this one: "Chatbots are a low-cost and highly scalable way to increase vaccinations". Yes, it could be, but it could also be the case that those recommendations are true only if the design follows the best practices from the behavioral literature. Using a chatbot does not necessarily guarantee success.

Additionally, the authors argue about the role of improved chatbots. However, there is growing evidence of potential backlash from users, particularly as chatbots become unrecognizable from humans. Therefore, the authors should recognize the potential limits of their recommendations.

3. Empirical analysis

Vaccination is a rare event in this sample. Only about 4% of the total sample gets vaccinated in the end. As such, the authors should use some alternative specifications to show the sensitivity of your findings to the modeling approach and the potential impact of the rare event issue. For example, they could use other specialized techniques for handling rare events in binary data, such as Firth's penalized likelihood approach or King and Zeng's correction for logistic regression.

4. Minor comments

Use of the expression "Two-way interactive chatbots". Oxford Languages defines Chatbot as "a computer program designed to simulate conversation with human users, especially over the internet." IBM defines it as "A chatbot is a computer program that simulates human conversation with an end user." Oracle defines "At the most basic level, a chatbot is a computer program that simulates and processes human conversation (either written or spoken), allowing humans to interact with digital devices as if they were communicating with a real person." It has to be clear that the authors are using a more rudimentary version of a chatbot that does not allow independent inputs from users.

External validity: How well does your sample represent the Chaco population?

Edit language: The document should be thoroughly edited. Many sentences are awkwardly constructed and complicated to read.

Reviewer #2:

Remarks to the Author:

In this ms. the Authors test whether a chatbot can get people to get a COVID booster shot, by comparison with a control condition with no communication, and a condition in which a reminder that they should get the booster is sent by text. The chatbot mostly consists in providing people with information about how to plan the booster shot (where to get it, etc.), and then a reminder before the appointment.

The effects of the chatbot are both very large (by comparison with the other conditions), and quite small (in absolute terms). However, we have to keep in mind that such interventions are bound to have small effects (few people click on links sent to them by phones), but that these (relatively) small effects are compensated by the low cost (by comparison, an ad campaign for instance might be both vastly more expensive, and even less effective).

This is a very solid ms. It's extremely rare to have access to vaccination data with that level of granularity (i.e. individual level data). It allows for a very strong experimental design. It's also great to be able to track exactly at what stage of the process people give up. Given the wide number of situations in which chatbots such as this one could be implemented, and the low cost of doing so, I think that the current results should be relevant to practitioners in many different fields (and countries, since this can now be implemented nearly everywhere in the world). As a result, I recommend acceptance, and I only have a few very minor comments.

Reading the ms., I had questions that might be useful for people attempting to implement a similar chatbot in other contexts. In particular, how much did it cost? Or maybe that measure would be too variable from one country to the next to be relevant? Also, was the chatbot implemented at scale after the experiment? It worked well, so one hopes that the chatbot was sent to all the inhabitants of the province, or even the country. Was that the case? If yes, and if the data is available, it would be interesting to see whether a spike in vaccination rates was apparent after the rollout (obvious it's not an experiment, so we couldn't infer causality, but still).

In the literature review, the Authors might consider adding

Altay, S., Hacquin, A., Chevallier, C. , & Mercier, H . Information delivered by a chatbot has a positive impact on COVID-19 vaccines attitudes and intentions. *Journal of Experimental Psychology: Applied*

Since this one is directly relevant to the topic of the paper (see also, on a different topic, but with a solid design, Altay, S., Schwartz, M., Hacquin, AS., Allard, A., Blancke, S. & Mercier, H. Scaling up interactive argumentation by providing counterarguments with a chatbot. *Nature Human Behavior*).

Figure 2: I think that *** is typically used for $p < .001$ and ** for $p < .01$ (otherwise, what is ** for, since the next threshold is $p < .05$, which is * right?).

In Figure 4, it would be good to have one last bar with the number of people who actually got the vaccine, so we can really follow all the steps of the process.

Reviewer #3:

Remarks to the Author:

In this study, the authors conducted a randomized, controlled trial among nearly 250,000 people in Argentina to evaluate impact of a two-way chatbot vs. one-way reminders vs. control to increase Covid vaccination boosters. This is an important topic and its findings could have implications for encourage other types of vaccinations and other health behaviors. The statistical analyses were well-described and conducted including correction for multiple comparisons. The paper could be improved by addressing the followings:

1. Abstract. The authors state the chatbot tripled vaccination compared to control and doubled it compared to the one-way reminders. It would be helpful to state the vaccination rates. It's unclear what the absolute impact was. A triple of a very small number is different than a tripling of a larger number. Moreover, the abstract does not state whether these changes were statistically significant or not.

2. Sample. It's important to note the population was restricted to those that had at least one prior dose of the Covid vaccine. That means the study is focused on Covid boosters and that the very vaccine resist individuals were removed. These two aspects of the sample are very important and not highlighted well in the abstract, paper, or discussion. It's really only stated in one sentence. It would be helpful for the reader if this was more clear.

3. Age. It would be helpful to know if the distribution of the sample in age was similar to that the country? It seemed to be skewed younger. It would be helpful to know the % age 65 or other rather than bundling them into the 50 years and over group.

4. Control vaccination rate was 0.70%. This is very low. The authors note the study occurred during a point of low infections. This is a significant limitation of the study. The authors should add a paragraph in the Discussions section on Limitations of the study and note this, as well as the restriction to individuals with a prior covid vaccination.

Author Rebuttal to Initial comments

Reviewer 1 comments

1. Main comment.

The RCT is well done, and I commend the authors for their work and for working with the government of a relatively poor province in Argentina.

My main comment has to do with their interpretation of the results. Basically, the authors argue that "We found that the chatbot more than tripled COVID-19 vaccine uptake, and nearly doubled uptake compared to the one-way message reminder." However, is it true that the authors can separate the effect of the chatbot from the content of the information?

There seems to be the case that two things differ from one treatment to the other: the interface (that is, having the ability to select options or not) and the content of the information (that is, those using the chatbot received more and better-tailored information than the others). The authors do not provide evidence that would allow the reader to identify the driving source of the effects.

- a. It could be the case that the one-way messages just provided less information, and that information was less behaviorally informed than the two-way messages. Therefore, it could be the case that the effect is not of the chatbot of itself but of the combination of behaviorally informed interventions that were delivered through the chatbot. In other words, it could be the case that users assigned to a one-way message receiving the same information would have acted the same.
- b. Alternatively, it could well be that the option to click on the message is what drives people because they feel empowered, because they increase their trust in the government by seeing a more developed communication method, or because the development of the chatbot serves as a signal of how relevant vaccination may be.

This is a limitation of the RCT that the authors should recognize across the whole document -even perhaps by changing the title. In other words, the evidence indicates that a chatbot that helped to deliver a set of behaviorally informed messages increased vaccination. Whether it was the chatbot or the messages can't be identified with this exercise.

Relatedly, the authors also argue that "The effect of the chatbot is comparable to that of other 'nudges' or behaviorally informed interventions for adult vaccinations." Again, the chatbot is a mechanism to deliver a behaviorally informed intervention. The chatbot allows the authors to personalize the message, to have a call for action, and to send a reminder. These are all standard 'nudges' that could have been introduced through alternative technologies.

We agree with Reviewer 1 that we are not able to distinguish whether it is the chatbot itself, as opposed to the set of behavioural tools that a chatbot enables, which causes the increase in vaccination. We have therefore adjusted the interpretation throughout the paper to explain that our results show that a chatbot which incorporates behaviourally-informed functionalities (which we now also refer to as a behaviourally-informed chatbot throughout the manuscript) performs better than a one-way message. We thank the

reviewer for the opportunity to clarify this.

We believe that this is still an important result, since it is very difficult or in some cases not possible to build these functionalities into a one-way message. In other words, a key advantage of a chatbot is that it enables you to incorporate behaviourally-informed functionalities in communications, and so testing whether the package of chatbot + behaviourally-informed functionalities outperforms a one-way message is still valuable for policymakers.

More specifically, we have made the following edits to the manuscript:

- We have changed the title to: *“Increasing vaccination: reminders work, a behaviourally-informed chatbot works better”*
- We have changed the abstract to clarify that our paper compares a chatbot service which incorporates behaviourally-informed functionalities to a one-way message: *“We designed and tested a new chatbot service to understand whether two-way interactive messaging incorporating behaviourally-informed functionalities could perform better than one-way message reminders.”*
- We refer to the chatbot as a *“behaviourally-informed chatbot”* throughout.
- We make references throughout the paper to the fact that the main treatment is a chatbot which incorporates behaviourally-informed functionalities including, for example, the following sentences:
 - *“The interactive nature of the chatbot allowed us to incorporate a set of behavioural interventions to promote vaccination which are not feasible in a one-way message.”*
 - *“Whilst there is a strong evidence base demonstrating that one-way text message reminders can increase vaccine uptake, no studies have tested whether an interactive chatbot incorporating a set of behavioural tools can do better.”*
 - *“By comparing the chatbot trial arm with the one-way message trial arm, we can understand whether a two-way interaction which incorporates behavioural functionalities causes a greater increase in vaccination rates than traditional one-way communications.”*
 - *“Our study shows that by going beyond simple information provision, a behaviourally-informed chatbot had nearly double the impact of a one-way message reminder on vaccine uptake.”*
- We also clarify the interpretation in the results section as follows:

- *“Unlike the one-way message, the chatbot offered both behaviourally-informed functionalities and the option to communicate*

two-way. Whilst we cannot test which of these aspects drove the effect of the chatbot, we know that the combination increased vaccination more than the traditional communications approach.”

2. Policy recommendations

The overall change in the interpretation of the results has a bearing on the policy recommendations. They should reconsider and qualify sentences like this one: `“\textit{Chatbots are a low-cost and highly scalable way to increase vaccinations}”`. Yes, it could be, but it could also be the case that those recommendations are true only if the design follows the best practices from the behavioral literature. Using a chatbot does not necessarily guarantee success.

Additionally, the authors argue about the role of improved chatbots. However, there is growing evidence of potential backlash from users, particularly as chatbots become unrecognizable from humans. Therefore, the authors should recognize the potential limits of their recommendations.

We agree with Reviewer 1, and we have rewritten the policy recommendations to make it clear that the results suggest that using a chatbot with behaviourally-informed tools is a low-cost and scalable way to increase vaccination. We have done this through the following sentences:

- *“Our three-arm randomised controlled trial has both shown that chatbots embedded with behavioural tools can work and that they can work better than one-way message reminders.”*
- *“Our study suggests that well-designed chatbots which incorporate behavioural functionalities can work, and may be a promising tool to increase the uptake of routine vaccinations for governments around the world.”*

We aren’t aware of academic evidence that chatbots cause backlash as they become unrecognisable from humans. Nonetheless, we have acknowledged the possibility that chatbots could be considered less trustworthy than traditional information sources in the policy recommendations as follows: *“Governments should nonetheless take care to ensure*

users consider their chatbot trustworthy before implementing at scale, given evidence that chatbots can be considered less trustworthy than traditional information sources in some sectors, such as financial services (57)”

3. Empirical analysis

Vaccination is a rare event in this sample. Only about 4% of the total sample gets vaccinated in the end. As such, the authors should use some alternative specifications to show the sensitivity of your findings to the modeling approach and the potential impact of the rare event issue. For example, they could use other specialized techniques for handling rare events in binary data, such as Firth's penalized likelihood approach or King and Zeng's correction for logistic regression.

We have conducted additional analyses using the two suggested methods (Firth's penalized likelihood approach and King and Zeng's correction for logistic regression) and reported the results in an updated version of Table S5 (the robustness checks table) in columns (5) and (6). The results are virtually identical to the primary regression. We describe those results in the robustness section of the supplementary information, as follows: *“To correct for rare events in binary data, in columns (5) and (6) we use King and Zeng's correction for logistic regression and Firth's penalised likelihood approach, respectively. Again, the results are almost identical.”*

4. Minor comments

Use of the expression “Two-way interactive chatbots”. Oxford Languages defines Chatbot as “a computer program designed to simulate conversation with human users, especially over the internet.” IBM defines it as “A chatbot is a computer program that simulates human conversation with an end user.” Oracle defines “At the most basic level, a chatbot is a computer program that simulates and processes human conversation (either written or spoken), allowing humans to interact with digital devices as if they were communicating with a real person.” It has to be clear that the authors are using a more rudimentary version of a chatbot that does not allow independent inputs from users.

We have added a sentence to the description of the chatbot to clarify that to operate the chatbot users choose from a set of options: *“In each case, users could reply to the chatbot's messages by choosing an answer from a set menu of options”.*

We have also added a footnote which provides our definition of a chatbot, following the academic literature: *“We define a chatbot as a computer program that simulates a conversation through two-way communication with human users (Adamopolou & Moussiades, 2020). “While chatbots simulate written conversation through two-way messaging capabilities, they are often distinguishable from communicating with a real person given the interaction format.” Adamopoulou, E. & Moussiades, L. (2020). An Overview of Chatbot Technology. Artificial Intelligence Applications and Innovations; 584:373–83. doi: 10.1007/978-3-030-49186-4_31.”*

External validity: How well does your sample represent the Chaco population?

We have added the following two sentences to explain how our sample compares to the population of Chaco in terms of both age and sex, based on data from the 2022 census: *“Compared to the adult population in Chaco, which is 47.6% male and 52.4% female, with 29.3% aged 18-29, 39.5% aged 30-49 and 31.2% aged 50 or older, our sample skews younger (40). This is because older people were more likely to have completed their vaccination schedule when we started the trial and therefore did not meet the inclusion criteria.”*

Edit language: The document should be thoroughly edited. Many sentences are awkwardly constructed and complicated to read.

We have made additional edits throughout the paper to improve how the sentences read (we have not listed these explicitly here as there are many and they sometimes involve small tweaks).

Reviewer 2 comments

Reading the ms., I had questions that might be useful for people attempting to implement a similar chatbot in other contexts. In particular, how much did it cost? Or maybe that measure would be too variable from one country to the next to be relevant? Also, was the chatbot implemented at scale after the experiment? It worked well, so one hopes that the chatbot was sent to all the inhabitants of the province, or even the country. Was that the case? If yes, and if the data is available, it would be interesting to see whether a spike in vaccination rates was apparent after the rollout (obvious it's not an experiment, so we couldn't infer causality, but still).

We have estimated the cost per additional vaccination, and mentioned this in the Discussion as follows: *“They can also do so at low cost, as we estimate that the chatbot would cost 46 USD per additional vaccination (55)”*.

In footnote 55, we have explained how we arrived at that figure: *“The intervention would cost 46 USD per additional vaccination. We arrive at this figure by first subtracting research costs from the total project costs. We divide the remaining implementation costs (177,000 USD, which includes staff salaries) by the number of additional vaccinations the chatbot would have caused had it been distributed to the full sample rather than withholding the chatbot from two trial arms for research purposes. Given that the chatbot caused an additional 1,300 vaccinations compared to the control group, we expect that it would have caused an additional 3,900 vaccinations had it been distributed to the full sample.”*

Whilst we wanted to scale the chatbot following the trial we have so far been unable to, owing to a change in government authorities. However, we are currently trying to implement a similar chatbot in another province.

In the literature review, the Authors might consider adding

Altay, S., Hacquin, A., Chevallier, C. , & Mercier, H . Information delivered by a chatbot has a positive impact on COVID-19 vaccines attitudes and intentions. *Journal of Experimental Psychology: Applied*

Since this one is directly relevant to the topic of the paper (see also, on a different topic, but with a solid design, Altay, S., Schwartz, M., Hacquin, AS., Allard, A., Blancke, S. & Mercier, H. Scaling up interactive argumentation by providing counterarguments with a chatbot. *Nature Human Behavior*).

We have added a citation in the introduction for: Altay, S., Hacquin, A., Chevallier, C. , & Mercier, H . Information delivered by a chatbot has a positive impact on COVID-19 vaccines attitudes and intentions. *Journal of Experimental Psychology: Applied*, as follows: *“This includes one study in a laboratory setting showing that information provision in a chatbot may increase the self-reported intention to receive the COVID-19 vaccine (33).”*

We agree that the second citation is not directly relevant to the topic of the paper and so

we have not included a reference to it.

Figure 2: I think that *** is typically used for $p < .001$ and ** for $p < .01$ (otherwise, what is ** for, since the next threshold is $p < .05$, which is * right?).

Our understanding is that *** for $p < 0.01$, ** for $p < 0.05$, and * for $p < 0.1$ is standard, and we have seen this in papers published at Nature Human Behaviour (e.g., here: https://static-content.springer.com/esm/art%3A10.1038%2Fs41562-024-01835-6/MediaObjects/41562_2024_1835_MOESM1_ESM.pdf). However, we are happy to change this if the editors would prefer to use the thresholds stated by Reviewer 2.

In Figure 4, it would be good to have one last bar with the number of people who actually got the vaccine, so we can really follow all the steps of the process.

We have created an updated version of Figure 4 where we have included an extra bar at the end (furthest right hand side) which states the number of people who got the vaccine in the chatbot trial arm.

Reviewer 3 comments

1. Abstract. The authors state the chatbot tripled vaccination compared to control and doubled it compared to the one-way reminders. It would be helpful to state the vaccination rates. It's unclear what the absolute impact was. A triple of a very small number is different than a tripling of a larger number. Moreover, the abstract does not state whether these changes were statistically significant or not.

We now state in the abstract that the two key effects were statistically significant, and have included the absolute effect size of the chatbot compared to the control group. The relevant sentences are: "*The behaviourally-informed chatbot more than tripled COVID-19 vaccine uptake, and nearly doubled uptake compared to the one-way message reminder. Both effects are statistically significant. In absolute terms, the chatbot caused a 1.6 percentage point increase in vaccinations compared to the control group.*"

2. Sample. It's important to note the population was restricted to those that had at least one prior dose of the Covid vaccine. That means the study is focused on Covid boosters and that the very vaccine resist individuals were removed. These two aspects of the sample are very important and not highlighted well in the abstract, paper, or discussion. It's really

only stated in one sentence. It would be helpful for the reader if this was more clear.

We have added a paragraph in the Discussion to highlight this limitation of the study:

“An important limitation of our study is that we do not know whether behaviourally-informed chatbots can increase vaccine uptake in vaccine resistant groups, as we excluded individuals who had not yet received their first dose of the COVID-19 vaccine from the sample. The chatbot was intended to make the process of getting a vaccination easier for those who already had at least some motivation to do so. It was not designed to try to address vaccine hesitancy or anti-vaccination attitudes.”

We have also expanded the paragraph in the introduction which explains this sample restriction to make it clearer (shortly after stating that we only include individuals who have already received at least one dose): *“The intervention intended to make the process of getting a vaccination easier for individuals who were not strongly vaccine-resistant”.*

3. Age. It would be helpful to know if the distribution of the sample in age was similar to that the country? It seemed to be skewed younger. It would be helpful to know the % age 65 or other rather than bundling them into the 50 years and over group.

We have added the following two sentences to explain how our sample compares to the population of Chaco in terms of both age and sex, based on data from the 2022 census.

Reviewer 2 is correct that it skews younger, which we have stated explicitly in the revision: *“Compared to the adult population in Chaco, which is 47.6% male and 52.4% female, with 29.3% aged 18-29, 39.5% aged 30-49 and 31.2% aged 50 or older, our sample skews younger (40). This is because older people were more likely to have completed their vaccination schedule when we started the trial and therefore did not meet the inclusion criteria.”*

We have not broken out the 65+ population explicitly in these descriptive statistics, because we believe it is most helpful to the reader to present age brackets that match the age brackets used as control variables in the regression analysis. However, we are happy to add this if the editors require it.

4. Control vaccination rate was 0.70%. This is very low. The authors note the study occurred during a point of low infections. This is a significant limitation of the study. The

authors should add a paragraph in the Discussions section on Limitations of the study and note this, as well as the restriction to individuals with a prior covid vaccination.

We have added a paragraph in the Discussions section to emphasize that the intervention was implemented at a point of low infections and low vaccine demand. We do not believe that this is a limitation of the study. When risk perception and susceptibility to infection diminish, people relax their preventive health behaviours. It is an important public health problem to understand how to induce preventive health behaviours during low-risk or perceived low-risk contexts, and our study shows that behaviourally-informed chatbots are a promising way to do this. We have therefore added the following:

“The behaviourally-informed chatbot was effective even at a time when demand for vaccination was low. Risk perception and susceptibility to infection are important motivators to get vaccinated (52-54). Identifying scalable interventions to sustain preventive behaviours when people are not motivated by a high-risk context is an important public health problem. Our study suggests that well-designed chatbots which incorporate behavioural functionalities can work, and may be a promising tool to increase the uptake of routine vaccinations for governments around the world.”

Nevertheless, we have also acknowledged that the results may not generalise to high vaccine demand settings in the limitations section of the Discussion, as follows:

“We are also unsure to what extent these results will generalise to contexts where demand for vaccinations is high given that the chatbot launched at a time when perceived infection risk from COVID-19 was low.”

We have addressed the limitation related to the restriction to individuals with a prior COVID-19 vaccination as outlined above under comment (2) by Reviewer 3 (through a limitations paragraph that we have added to the Discussion).

Decision Letter, first revision:

13th June 2024

Dear Dr. Brown,

Thank you for your patience as we've prepared the guidelines for final submission of your Nature

Human Behaviour manuscript, "Increasing vaccination: reminders work, a behaviourally-informed chatbot works better" (NATHUMBEHAV-23113997A). Please carefully follow the step-by-step instructions provided in the attached file, and add a response in each row of the table to indicate the changes that you have made. Please also address the additional marked-up edits we have proposed within the reporting summary. Ensuring that each point is addressed will help to ensure that your revised manuscript can be swiftly handed over to our production team.

We would hope to receive your revised paper, with all of the requested files and forms within two-three weeks. Please get in contact with us if you anticipate delays.

Nature Human Behaviour offers a Transparent Peer Review option for new original research manuscripts submitted after December 1st, 2019. As part of this initiative, we encourage our authors to support increased transparency into the peer review process by agreeing to have the reviewer comments, author rebuttal letters, and editorial decision letters published as a Supplementary item. When you submit your final files please clearly state in your cover letter whether or not you would like to participate in this initiative. Please note that failure to state your preference will result in delays in accepting your manuscript for publication.

In recognition of the time and expertise our reviewers provide to Nature Human Behaviour's editorial process, we would like to formally acknowledge their contribution to the external peer review of your manuscript entitled "Increasing vaccination: reminders work, a behaviourally-informed chatbot works better". For those reviewers who give their assent, we will be publishing their names alongside the published article.

Cover suggestions

We welcome submissions of artwork for consideration for our cover. For more information, please see our guide for cover artwork.

ORCID

Non-corresponding authors do not have to link their ORCIDs but are encouraged to do so. Please note that it will not be possible to add/modify ORCIDs at proof. Thus, please let your co-authors know that if they wish to have their ORCID added to the paper they must follow the procedure described in the

following link prior to acceptance: <https://www.springernature.com/gp/researchers/orcid/orcid-for-nature-research>

Nature Human Behaviour has now transitioned to a unified Rights Collection system which will allow our Author Services team to quickly and easily collect the rights and permissions required to publish your work. Approximately 10 days after your paper is formally accepted, you will receive an email in providing you with a link to complete the grant of rights. If your paper is eligible for Open Access, our Author Services team will also be in touch regarding any additional information that may be required to arrange payment for your article.

Please note that *Nature Human Behaviour* is a Transformative Journal (TJ). Authors may publish their research with us through the traditional subscription access route or make their paper immediately open access through payment of an article-processing charge (APC). Authors will not be required to make a final decision about access to their article until it has been accepted. Find out more about Transformative Journals

[REDACTED]

Best regards,
[REDACTED]

On behalf of
[REDACTED]

Reviewer #1:
Remarks to the Author:
No further comments

Reviewer #2:

Remarks to the Author:

I'm happy with the changes made and recommend that the paper be published.

Final Decision Letter:

Dear Dan,

I am happy to let you know that your Article "A behaviourally-informed chatbot increases vaccination rates in Argentina more than a one-way reminder", has now been accepted for publication in *Nature Human Behaviour*.

Please note that *Nature Human Behaviour* is a Transformative Journal (TJ). Authors may publish their research with us through the traditional subscription access route or make their paper immediately open access through payment of an article-processing charge (APC). Authors will not be required to make a final decision about access to their article until it has been accepted. Find out more about Transformative Journals

If you have posted a preprint on any preprint server, please ensure that the preprint details are updated

with a publication reference, including the DOI and a URL to the published version of the article on the journal website.

With best regards,
[REDACTED]